# Impact of the ^18^F-FDG-PET/MRI on Metastatic Staging in Patients with Hepatocellular Carcinoma: Initial Results from 104 Patients

**DOI:** 10.3390/jcm10174017

**Published:** 2021-09-06

**Authors:** Mathilde Vermersch, Sébastien Mulé, Julia Chalaye, Athena Galletto Pregliasco, Berivan Emsen, Giuliana Amaddeo, Aurélien Monnet, Alto Stemmer, Laurence Baranes, Alexis Laurent, Vincent Leroy, Emmanuel Itti, Alain Luciani

**Affiliations:** 1Medical Imaging Department, Henri Mondor Hospital, APHP, 94000 Créteil, France; sebastien.mule@aphp.fr (S.M.); athena.gallettopregliasco@aphp.fr (A.G.P.); laurence.baranes@aphp.fr (L.B.); alain.luciani@aphp.fr (A.L.); 2Institut Mondor de la Recherche Biomédicale (IMRB) Team 18, INSERM Unit 955, Henri Mondor Hospital, 94000 Créteil, France; 3Medical Imaging Department, Lille University Hospital, 59000 Lille, France; 4Nuclear Medicine Department, Henri Mondor Hospital, APHP, 94000 Créteil, France; julia.chalaye@aphp.fr (J.C.); berivan.emsen@gmail.com (B.E.); emmanuel.itti@aphp.fr (E.I.); 5Department of Hepatogastroenterology, Henri Mondor Hospital, APHP, 94000 Créteil, France; giuliana.amaddeo@aphp.fr (G.A.); vincent.leroy2@aphp.fr (V.L.); 6Siemens Healthineers, Siemens Healthcare GmbH, 91052 Erlangen, Germany; aurelien.monnet@siemens-healthineers.com (A.M.); alto.stemmer@siemens-healthineers.com (A.S.); 7Hepatobiliary Surgery and Liver Transplantation, Henri Mondor Hospital, APHP, 94000 Créteil, France; alexis.laurent@aphp.fr

**Keywords:** hepatocellular carcinoma, staging, metastases, PET/MRI, ^18^F-Fluoro-Desoxy-Glucose

## Abstract

Optimal HCC therapeutic management relies on accurate tumor staging. Our aim was to assess the impact of ^18^F-FDG-WB-PET/MRI on HCC metastatic staging, compared with the standard of care CT-CAP/liver MRI combination, in patients with HCC referred on a curative intent or before transarterial radioembolization. One hundred and four consecutive patients followed for HCC were retrospectively included. The WB-PET/MRI was compared with the standard of care CT-CAP/liver MRI combination for HCC metastatic staging, with pathology, followup, and multidisciplinary board assessment as a reference standard. Thirty metastases were identified within 14 metastatic sites in 11 patients. The sensitivity of WB-PET/MRI for metastatic sites and metastatic patients was significantly higher than that of the CT-CAP/liver MRI combination (respectively 100% vs. 43%, *p* = 0.002; and 100% vs. 45%, *p* = 0.01). Metastatic sites missed by CT-CAP were bone (*n* = 5) and distant lymph node (*n* = 3) in BCLC C patients. For the remaining 93 nonmetastatic patients, three BCLC A patients identified as potentially metastatic on the CT-CAP/liver MRI combination were correctly ruled out with the WB-PET/MRI without significant increase in specificity (100% vs. 97%; *p* = 0.25). The WB-PET/MRI may improve HCC metastatic staging and could be performed as a “one-stop-shop” examination for HCC staging with a significant impact on therapeutic management in about 10% of patients especially in locally advanced HCC.

## 1. Introduction

Hepatocellular carcinoma (HCC) is the most common primary liver cancer, with the sixth highest incidence rate worldwide, and the second leading cause of cancer-related death [1]. In the setting of a cirrhotic liver, HCC can be diagnosed by noninvasive imaging [2,3,4]. Liver magnetic resonance imaging (MRI) is routinely advocated for locoregional staging owing to its higher sensitivity over computed tomography (CT) for liver nodule detection and characterization [5]. For metastatic staging, chest–abdomen–pelvis CT (CT-CAP) remains the standard of care because of its availability and higher spatial resolution. The majority of patients are therefore treated after diagnosis and staging made by the CT-CAP/liver MRI combination using the Barcelona Clinic Liver Cancer (BCLC) standard of care recommendations [2,3,4].

Patients eligible for curative therapies (resection, transplantation, or local ablation), without extrahepatic extension, have a median survival rate above 60 months [6]. On the other hand, locoregional therapies, including the recently introduced transarterial radioembolization (TARE) technique, remain inappropriate in patients with advanced HCC especially those having an extrahepatic tumor, leading to survival rates below 18% at 5 years [2,7,8]. As a result, improving the detection of extrahepatic disease on imaging is of interest both in early as well as in locally advanced HCC.

A whole-body MRI with diffusion weight imaging (DWI) sequences appears to be a promising whole-body technique for the assessment of distant metastases in patients with malignant tumors [9,10,11], especially regarding the early identification of bone involvement. Similarly, ^18F^FluoroDesoxyGlucose (^18^F-FDG PET/CT) optimizes HCC staging and therapeutic management [12,13]. It upgrades metastatic staging, especially for lymph node involvement and bone metastases [14,15], improves treatment allocation for patients on the waiting list for liver transplantation [16], and enables a response assessment after TARE [17,18]. Whole-body ^18^F-FDG PET/MRI (WB-PET/MRI) is an emerging hybrid modality, which appears highly promising in oncology settings [19,20,21]. PET/MRI theoretically provides both optimal local staging by liver MRI, metastatic staging with WB-MRI potentially equivalent to or better than that of standard technologies, and ^18F^FDG-PET prognostic information in a one-stop-shop imaging approach. Moreover, a major advantage of PET/MRI is the absence of X-ray radiation, especially in an oncological setting with repeated examinations.

The aim of our study was, therefore, to assess the impact of the WB-PET/MRI on HCC metastatic staging, compared with the standard of care CT-CAP/liver MRI combination, in patients with HCC referred on a curative intent (before surgery, local ablation, or liver transplantation) or before TARE.

## 2. Materials and Methods

### 2.1. Patient Population

All patients aged 18 years or older referred to our institution for HCC and who underwent WB-PET/MRI between 26 June 2017 and 18 December 2019 were eligible for this retrospective institutional review board (IRB)-approved study (IRB number, CRM-1910-032). Written informed consent was waived by IRB. A total of 141 patients were retrospectively screened for inclusion. Inclusion criteria were as follows: (i) HCC patients with BCLC stage A, B or C, (ii) WB-PET/MRI baseline evaluation before curative therapies (including orthotopic liver transplantation, surgical resection, and local ablation) or before TARE, and (iii) CT-CAP performed less than three months before or after the WB-PET/MRI and without any therapy performed in the meantime. A total of 120 patients were subsequently included. Eighteen of these were excluded from analysis because of incomplete WB-PET/MRI protocol (*n* = 9) or because pathological diagnosis revealed cholangiocarcinoma (*n* = 6) or epithelioid angiomyolipoma (*n* = 1). As a result, the final study population consisted of 104 patients (Figure 1).

### 2.2. Clinical and Biological Data

The following patients’ characteristics were collected retrospectively from the electronic medical records: gender, age, underlying liver disease and its etiology, presence of cirrhosis, Child–Pugh score, the Eastern Cooperative Oncology Group (ECOG) performance status, and alpha Fetoprotein level. The number of patients referred to WB-PET/MRI on a curative intent or prior to TARE was assessed.

### 2.3. Imaging Protocol

All patients underwent WB-PET/MRI on an integrated Biograph mMR 3T scanner (Siemens Healthineers, Erlangen, Germany). After fasting for at least 6 h to ensure normal blood glucose levels, patients were injected with an average 4.3 MBq/kg ^18^F-FDG, and images were acquired 1 h later.

First, a whole-body MRI was performed over five stations of 3 min duration each from the top of the head to the knees (ensuring that one station covered the entire liver) with simultaneous acquisition of PET using HD-Chest^®^ motion correction (a triggering technique collecting counts during the portion of the respiratory cycle with the least motion) and two MR sequences: a 3D-T1-Dixon with Controlled Aliasing in Parallel Imaging Results in Higher Acceleration (CAIPIRINHA) and a blipped Simultaneous-Multi-Slice Diffusion-Weighted-Imaging (SMS-DWI) sequence with 3 b values (50, 400, and 800 s/mm^2^). The 3D-T1 Dixon generated the µ-maps using a 5-compartment segmentation of air, fat, muscles, lungs, and bones, and PET images were reconstructed with and without MR attenuation correction (MRAC) using an iterative 3D Ordered Subset Expectation Maximization algorithm with point-spread function modeling in 344 × 344 matrices.

Secondly, a dedicated liver station was acquired with simultaneous acquisition of PET using BodyCOMPASS^®^ (Siemens Healthineers, Erlangen, Germany) motion management (a triggering technique collecting counts during the entire respiratory cycle in five bins and applying deformation fields on the different bins to elastically match the one with the least motion) and the following MR sequences: a 3D-T1-Dixon with CAIPIRINHA, an axial T2-weighted HASTE, and a blipped SMS-DWI IntraVoxel Incoherent Motion (SMS-IVIM-DWI) sequence with 10 b values (0, 10, 20, 30, 50, 80, 100, 200, 400 and 800 s/mm^2^). Duration of this dedicated liver step was 15 to 18 min. Afterward, a dynamic contrast-enhanced 3D-T1-GRE-Dixon-VIBE-CAIPIRINHA was acquired after injection of gadolinium–chelate including two late arterial phases, one portal-venous phase, and one delayed phase. These specific sequences associated with liver stations of 3D-T1-MRAC and SMS-DWI were defined as the conventional liver MRI sequences. Whole-body axial T1-GRE-Dixon-VIBE-CAIPIRINHA postinjection and chest centered axial T1-GRE-StarVIBE (sequence with non-Cartesian acquisition of k-space) ended the protocol and were acquired at least 4 min after injection of gadolinium–chelate. The SMS sequences used in this study (blipped SMS-DWI, SMS-IVIM-DWI) are prototypes (not commercially available).

The median total acquisition time was 79 min (range, 57–134 min). WB-PET/MRI protocol is detailed in Appendix A.

All patients underwent a multiphasic contrast-enhanced CT-CAP scan including at least portal venous phase images covering the chest, abdomen, and pelvis. Images were automatically acquired 70 s after contrast material administration. Acquisition parameters were as follows: tube voltage, 120 kVp; tube current-time, 277 mAs; section collimation, 64 × 1.25 mm; scan time per spiral, 0.5 s. The median time between the CT-CAP and WB-PET/MRI was 14 days (range, 0–90 days).

### 2.4. Imaging Evaluation

Examinations were separated into two whole-body datasets: (1) CT-CAP associated with liver MRI which constitutes the standard of care protocol and (2) WB-PET/MRI.

The CT-CAP, liver MRI, and whole-body MRI were evaluated by a radiologist experienced in liver imaging, and whole-body PET and liver centered PET acquisitions were analyzed by a nuclear medicine physician experienced in HCC evaluation. The WB-PET/MRI was evaluated by consensus of both readers in a second reading session performed four weeks after each individual analysis. Readers were aware of the clinical history but were blinded to any prior imaging and to the other imaging dataset.

For each patient, the following individual metastatic sites including lung, bone, and additional metastatic sites were systematically assessed. For analysis, a 4-grade scale was developed to categorize sites according to their probability of metastatic involvement: 1 = benign lesion, 2 = probably benign lesion — to be controlled on followup, 3 = probably malignant lesion — needing further investigation or close followup, and 4 = definitely malignant lesion. An individual grade was attributed separately for the two image datasets (CT-CAP/liver MRI combination and WB-PET/MRI) accounting for the probability of metastasis on a per patient basis. Patients with grade-3 and grade-4 lesions were considered as metastatic.

For each metastatic site, the number of metastatic lesions were determined whether on the CT-CAP/liver MRI or WB-PET/MRI datasets.

Last, the total number of metastatic patients, metastatic sites and metastatic lesions was assessed for both datasets.

### 2.5. Reference Standard

Pathology, followup, and multidisciplinary board assessment using all data available were used as the reference standard to define metastatic involvement in each patient. If pathology was not available, followup images were used as the standard of reference. When lesions remained stable on 6-month followup images they were considered as being without metastasis. If lesions showed a size or number increase on subsequent images they were considered metastases. The median followup time was nine months (range, 0–30 months). Followup was not available (<3 months) for nine patients because of death (*n* = 5), or lost to followup (*n* = 4).

### 2.6. Endpoint

The main endpoint was the HCC metastatic staging (M-staging). The reference standard was used to evaluate the sensitivity and specificity of the two imaging datasets in determining M-staging on a per-lesion, per-site, and per-patient analysis.

### 2.7. Statistical Analyses

Categorical data are represented as numbers (percentage). Continuous variables are represented as median (range).

Diagnostic performance and diagnostic confidence of each dataset were evaluated on a per-lesion, per-site, and per-patient basis and compared (exact Fisher’s test or Chi-2 as appropriate).

The number of patients for whom WB-PET/MRI led to any theoretical change in overall staging or in the therapeutic management was collected.

All statistical analyses were performed using SPSS version 23; a *p*-value < 0.05 was considered as significant.

## 3. Results

### 3.1. Population Characteristics

Eighty-eight men and sixteen women with a median age of 63 years (range, 35–86 years) were included. One hundred patients (100/104; 96%) had chronic liver disease (83/104 at cirrhotic stage; 80%). Half of the liver diseases were linked to a viral etiology (52/104, 50%).

A total of 67 patients were referred for WB-PET/MRI on a curative intent, whether prior to surgery or local ablation workup(33 patients), including 26 BCLC Stage A, and 7 BCLC stage B patients, or prior to liver transplantation(34 patients) including 10, 13, and 11 patients at of BCLC stages A/B/C respectively.

Thirty-seven patients were referred for WB-PET/MRI prior to TARE (including 2, 6 and 29 patients at of BCLC stage A/B/C respectively). Patients’ characteristics are detailed in Table 1.

The M-staging was positive in 11 (10.5%) patients according to the reference standard, with a total of 30 individual metastases identified in 14 metastatic sites.

### 3.2. Diagnostic Accuracy of WB-PET/MRI for Metastatic Staging

The WB-PET/MRI enabled the correct identification of all metastatic and nonmetastatic patients, resulting in a sensitivity and a specificity of 100% (11/11 and 93/93 patients, respectively). The standard of care CT-CAP/liver MRI combination had a significantly lower sensitivity (5 of 11 patients, 45%; *p* = 0.01) and a comparable specificity (90 of 93 patients, 97%; *p* = 0.25).

All 14 metastatic sites were correctly identified on the WB-PET/MRI, although only 7 of 16 individual pulmonary metastases (44%) were seen (Figure 2). On the CT-CAP/liver MRI dataset, six metastatic sites were correctly identified (6/14; 43%), resulting in a significantly lower sensitivity (43% vs. 100%, *p* = 0.002). Notably, all 10 individual bone metastases were missed on the CT-CAP/liver MRI (Figure 3), as well as two mediastinal lymph nodes (Figure 4) and one retroperitoneal lymph node involvement (Figure 3).

The sensitivities of the two datasets and of the whole-body MRI or whole-body PET alone for M-staging on a per-lesion, per-site, and per-patient analysis are detailed in Table 2.

In metastatic patients, the mean individual grade accounting for the probability of metastasis was significantly higher when using the WB-PET/MRI over standard of care imaging (3.64 vs. 2.36; *p* = 0.005) suggesting a higher confidence in the metastatic nature of the lesions identified on the WB-PET/MRI.

### 3.3. Impact on Patient Management

The WB-PET/MRI lead to changes in therapeutic management in 10 of 104 patients (9.6%). Metastatic lesions were identified by the WB-PET/MRI in two BCLC stage C patients referred before liver transplantation (Figure 3) and in five BCLC stage C patients referred before TARE (Figure 2 and Figure 4).

In three BCLC stage A patients imaged before surgical resection, three lesions suspected of metastasis on the CT-CAP/liver MRI (Figure 5 and Figure 6) were correctly invalidated on the WB-PET/MRI owing to their hypometabolism on the PET.

The impact on patient management resulting from the WB-PET/MRI is detailed in Table 3.

## 4. Discussion

Accurate staging is crucial for appropriate treatment planning in patients with HCC. This study is the first to report the integration of the WB-PET/MRI into HCC management in 104 consecutive patients referred before curative treatment or expensive locoregional therapy (TARE). The results of this study suggest that the WB-PET/MRI is superior to the combined analysis of CT-CAP and liver MRI, which is currently the standard of care imaging strategy for HCC M-staging. In addition, the WB-PET/MRI led to changes in the therapeutic management in almost 10% of patients included in our study especially in BCLC C patients by detecting additional metastases or in BCLC A patients by reducing the uncertainty regarding metastatic involvement. We deliberately included patients involved in distinct therapeutic pathways whether on a curative intent or prior to TARE. Interestingly, both pathways were positively impacted by the use of the WB-PET/MRI. False positive lesions of the CT-CAP and liver MRI combination, which could delay treatment or lead to invasive additional diagnostic procedure, were correctly addressed by the WB-PET/MRI in early stage HCC patients. In addition, the WB-PET/MRI enabled the correct identification of metastases among patients with locally advanced HCC. This is of utmost importance both in patients eligible for liver transplantation but also for patients referred for TARE as in both situations, metastases dramatically impact individual survival [2,3,4].

The incidence of metastases in HCC patients is high involving three main extrahepatic locations, namely lung, lymph nodes, and bone [22]. Even if computed tomography is the method of choice for lung evaluation, its accuracy for lymph node involvement [23] and bone lesion [24] remains weak. In our study, the WB-PET/MRI allowed the detection of all patients with bone metastases and lymph node involvement whereas the CT-CAP and liver MRI combination missed all bone metastases and 75% of lymph node involvement. This is in agreement with literature data which highlight the potential of the PET/MRI in malignant musculoskeletal disease [25]. In addition, combining the WB-MRI with PET data favors the detection of bone lesions especially when compared to WB-PET/CT as is reported in breast cancer [26]; in our study, half of bone lesions were visible only on MRI sequences without visualization on PET. On the other hand, the CT allowed detection of all lung metastases whereas, the WB-PET/MRI sensitivity was 44% on a per-lesion basis. This finding is in agreement with literature data for lung metastases of less than 1 cm [27,28]. However, all patients with lung metastases were correctly detected by the WB-PET/MRI, suggesting that the WB-PET/MRI can indeed allow adequate patient screening for lung metastases. A recent article found similar results with the WB-PET/MRI showing a reduced sensitivity for lung nodule detection but without impact in clinical management [29]. Furthermore, in case of solitary lung nodules for which a CT scan does not enable confident characterization, functional information derived from DWI and metabolic information derived from the PET can help to determine the malignant potential [30].

With reference to metastatic sites and metastatic patients, we found a 100% accuracy for the WB-PET/MRI. Indeed, the combination of the WB-MRI and WB-PET improved the diagnostic performance compared with the CT-CAP and liver MRI combination, as infracentimetric lesions not visible on PET could be detected by high-resolution MRI, and undetermined lesions on MRI could be invalidated or confirmed on PET. Hence, the WB-MRI and PET appear complementary with the MRI providing a high sensitivity and the PET providing a high specificity, allowing the invalidation of potentially suspicious lesions. Third, even if the accuracy on a per-patient or per-site analysis was 100%, not all individual metastatic lesions were found by the WB-PET/MRI, especially lung metastases.

WB-PET/MRI examinations were performed using ^18^F-FDG and not ^18^F-Choline. Indeed, 18F-Choline is not available worldwide, and data from the literature highlight the impact of 18F-FDG for prognostic staging including metastatic assessment.

Some limitations of this pilot study should be mentioned. First, the study cohort was retrospectively selected and limited to a single institution, with a limited number of patients included, potentially leading to selection bias. Second, followup time was limited, with a median followup time of nine months, which may have led to nondetection of small slowly progressive metastases. However, the potential therapeutic and prognostic impact of a single small size lung nodule missed at imaging is not well established. Followup was not available for nine patients (including two metastatic patients). However, the two metastatic patients without followup died during the first three months after the WB-PET/MRI and both showed a significant rise in serum alpha-fetoprotein level together with significant uptake on ^18^F-Choline PET/CT during followup. Third, only 15 metastatic sites in 11 patients were present in our patient population. The small number of metastatic lesions and metastatic sites may have led to a lack of statistical power. Fourth, the diagnostic performance of WB-PET/MRI was not compared to that of the PET/CT; however, the current standard of care for HCC staging relies on the association of liver MRI and CAP-CT, while ^18^F-FDG PET/CT is not recommended by both EASL and AASLD [2,4]. Last, we did not compare the accuracy of local and regional staging between protocols because the reference standard for HCC local staging, liver MRI, was part of both the standard of care and WB-PET/MRI datasets.

## 5. Conclusions

The WB-PET/MRI can be performed as a “one-stop-shop” examination and may significantly improve the sensitivity and diagnostic confidence for metastatic staging in HCC patients. The WB-PET/MRI can impact patient management in up to 10% of patients, especially in locally advanced BCLC C patients by detecting additional metastases or in BCLC A patients by reducing the uncertainty about metastatic involvement. Prospective studies as well as cost/utility analyses are needed to best define the role of the WB-PET/MRI in HCC staging in clinical routine.

## Figures and Tables

**Figure 1 jcm-10-04017-f001:**
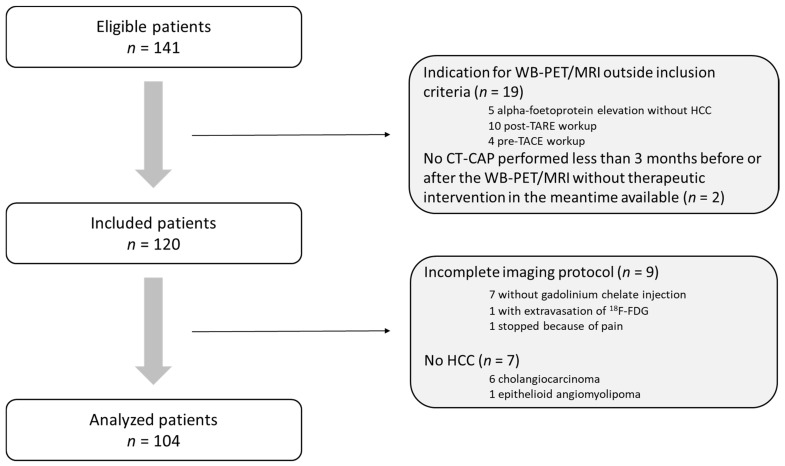
Flow Chart.

**Figure 2 jcm-10-04017-f002:**
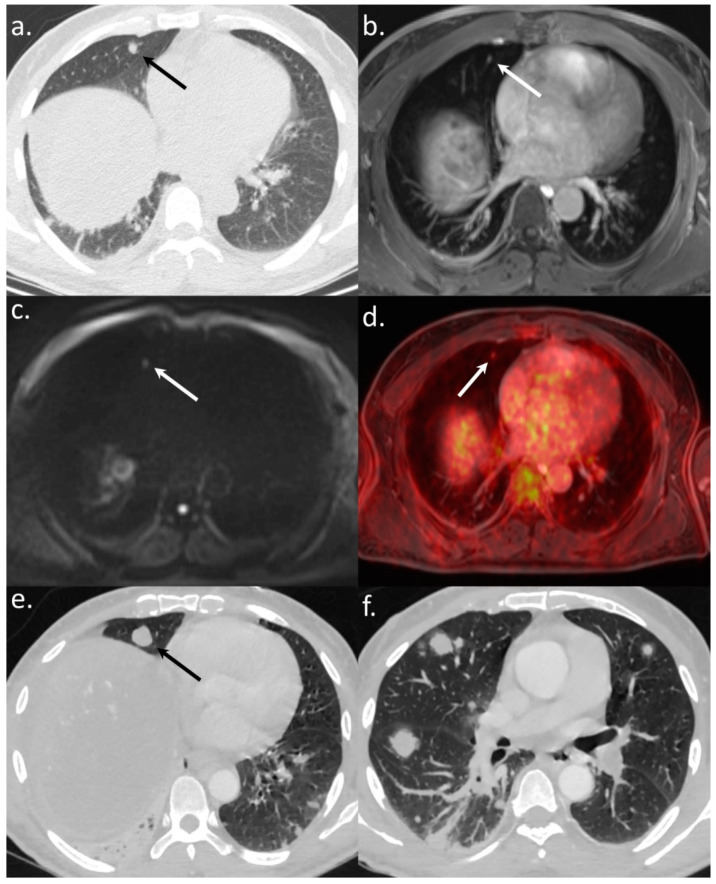
A 56-year-old patient referred to WB-PET/MRI before TARE. The CT-CAP (**a**) showed a solitary rounded 7 mm-size lung nodule (arrows). The WB-PET/MRI confirmed an infracentimetric lung nodule visible on morphologic sequences (T1-weighted after gadolinium injection) (**b**) with hyperintensity on b800 s/mm^2^ DWI (**c**) and ^18^F-FDG hypermetabolism (**d**) leading to the diagnosis of lung metastasis. The followup CT scan (**e**,**f**) performed three months after the WB-PET/MRI showed a significant progression of the number and size of lung nodules, confirming the metastatic status.

**Figure 3 jcm-10-04017-f003:**
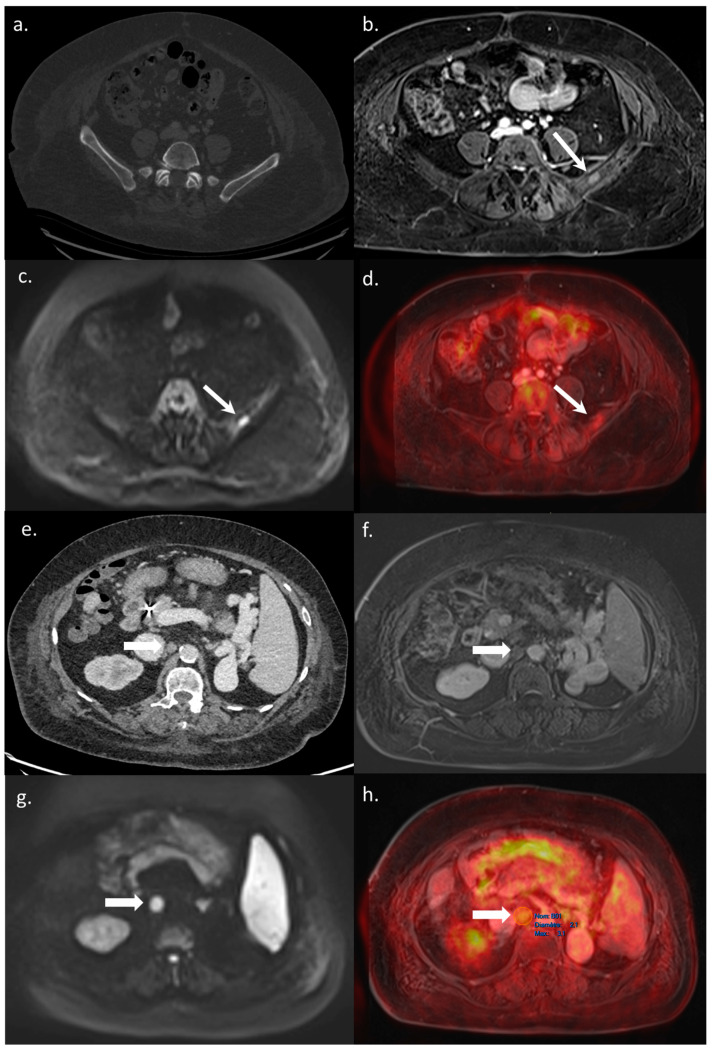
A 65-year-old BCLC C patient referred for a WB-PET/MRI before liver transplantation. On CT-CAP, no bone lesion was visible (**a**). The WB-PET/MRI revealed an infracentimetric bone lesion with enhancement after Gadolinium–chelate injection (**b**), together with hyperintensity on DWI (**c**), and focal hypermetabolism on ^18^F-FDG-PET (**d**) leading to the diagnosis of bone metastasis. Moreover, on CT-CAP, a retroperitoneal nonspecific lymph node was observed (**e**), visible on morphologic MRI sequence (**f**). Both diffusion restriction (**g**) and ^18^F-FDG hypermetabolism (**h**) were observed leading to the diagnosis of lymph node involvement. The patient was excluded from the liver transplantation list, with a rapidly progressing disease leading to patient death within three months.

**Figure 4 jcm-10-04017-f004:**
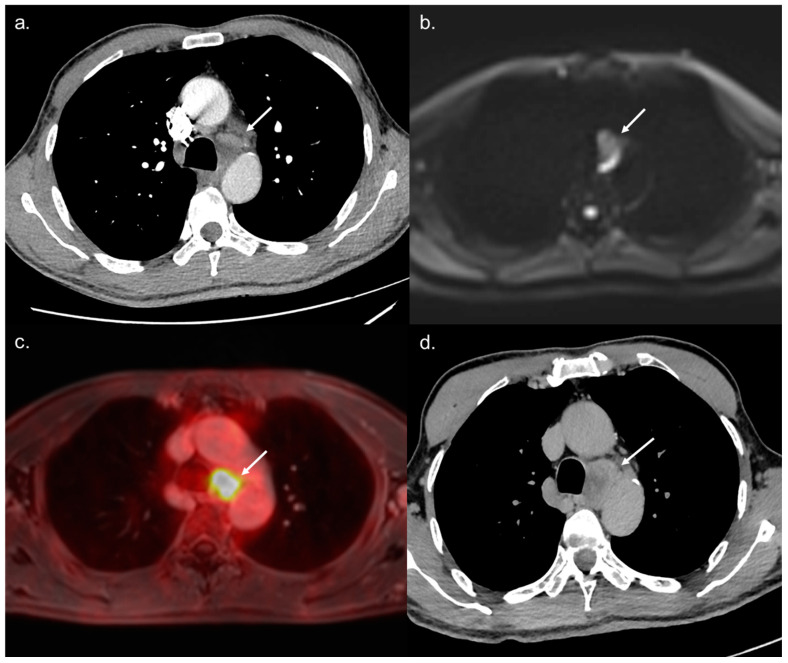
A 68-year-old BCLC C patient referred to WB-PET/MRI before TARE. The CT-CAP performed (**a**) showed a 12 × 25 mm large mediastinal lymph node of indeterminate nature. The WB-PET/MRI revealed hyperintensity on b800 s/mm^2^ DWI (**b**) with hypermetabolism on ^18^F-FDG-PET (**c**) suggestive of metastatic involvement. Followup performed two months later confirmed the metastatic nature of the lymph nodes with rapid increase in size and necrosis (**d**).

**Figure 5 jcm-10-04017-f005:**
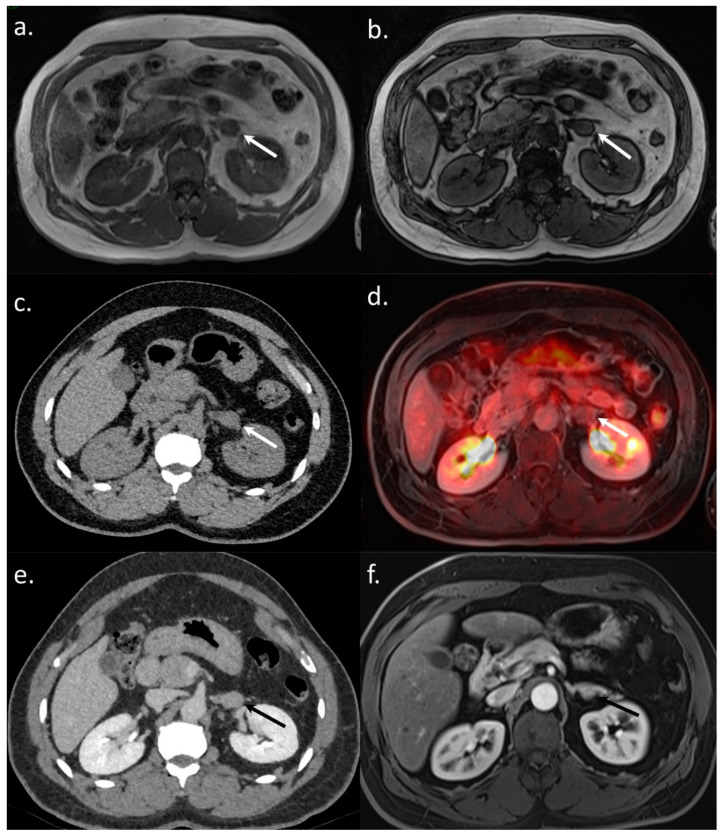
A 61-year-old BCLC A patient referred for a WB-PET/MRI before liver surgery for HCC. On the liver MRI, a 2 cm large left adrenal nodule was observed without signal drop on out phase images (**a**,**b**). On the CT-CAP, the unenhanced adrenal mass density was 39UH, which was not able to confirm its benign nature (**c**). No hypermetabolism was observed on ^18^F-FDG-PET (**d**), leading to the exclusion of its metastatic nature. Surgery was performed and the adrenal lesion remained stable on the followup CT (**e**) and MRI (**f**) performed three and nine months after the WB-PET/MRI, respectively.

**Figure 6 jcm-10-04017-f006:**
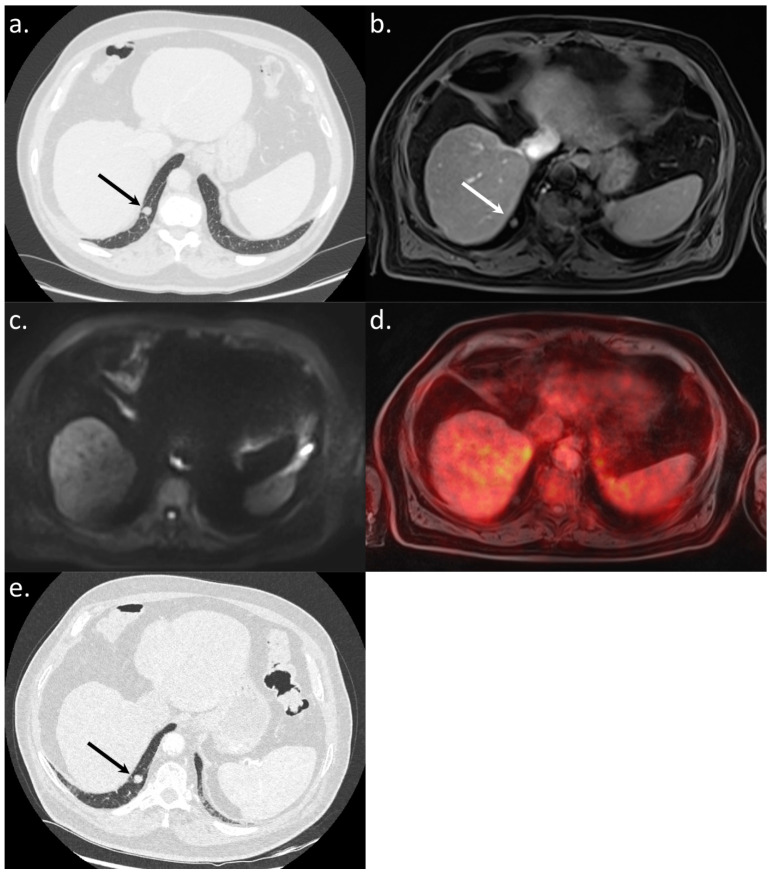
An 82-year-old BCLC A patient referred for a WB-PET/MRI before liver surgery for HCC. The CT-CAP showed an 11 mm large round-shaped pulmonary nodule (**a**). The nodule was also visible on the morphologic MRI sequence (**b**). There was no diffusion restriction (**c**) and no ^18^F-FDG hypermetabolism (**d**) leading to the exclusion of lung metastasis. Surgery was performed and, on CT followup performed two years after the WB-PET/MRI (**e**), the nodule remained stable without the appearance of additional lesions.

**Table 1 jcm-10-04017-t001:** Patients and tumor features including clinical and biological data.

Clinical and Biological Data	Subgroups	*n* (%)/Median [Range]
Gender	Men	88/104 (84.6)
Age		63 [35; 86]
Chronic liver disease		100/104 (96)
	HBV	23/104 (22)
	HCV	28/104 (27)
	HBV+HCV	1/104 (1)
	Excessive alcohol consumption	20/104 (19)
	NASH	6/104 (6)
	Excessive alcohol consumption + NASH	17/104 (16)
	Budd Chiari	2/104 (2)
	Auto immune	3/104 (3)
Cirrhosis		83/104 (80)
αFP serum level (ng/mL)		10 [1; 300,000]
Indication	Pre surgery or pre percutaneous ablation workup	33/104 (31.7)
	Pre liver transplantation Workup	34/104 (32.7)
	Pre TARE workup	37/104 (35.6)
BCLC (reference standard)	A	38/104 (36.5)
	B	26/104 (25)
	C	40/104 (38.5)

HBV: Hepatitis B virus; HCV: Hepatitis C Virus; NASH: nonalcoholic steato-hepatitis; αFP: alpha-fetoprotein; TACE: Transarterial-chemoembolization; TARE: Transarterial radioembolization; BCLC: Barcelona Clinic Liver Cancer.

**Table 2 jcm-10-04017-t002:** Per-lesion, per site, and per-patient sensitivities of the CT-CAP/liver MRI combination, WB-PET/MRI, MRI, and PET for distant metastases detection according to their localization.

		Reference Standard	CT-CAP/Liver MRI		WB-PET/MRI			MRI			PET		
			*n*	Sensitivity (%)	*n*	Sensitivity (%)	*p*-Value	*n*	Sensitivity (%)	*p*-Value	*n*	Sensitivity (%)	*p*-Value
Lung Metastases	Per-Lesion Analysis	16	16	100	7	44	0.0008	7	44	0.0008	6	38	0.0002
	Per-Patient Analysis	5	5	100	5	100	1	5	100	1	5	100	1
Bone Metastases	Per-Lesion Analysis	10	0	0	10	100	<0.001	10	100	<0.001	5	50	0.03
	Per-Patient Analysis	5	0	0	5	100	0.008	5	100	0.008	4	80	0.047
Other Metastases	Per-Lesion Analysis	4	1	25	4	100	0.14	4	100	0.14	4	100	0.14
	Per-Patient Analysis	4	1	25	4	100	0.14	4	100	0.14	4	100	0.14
Total	Per-Lesion Analysis	30	17	57	21	70	0.28	21	70	0.28	15	48	0.19
	Per-Site Analysis	14	6	43	14	100	0.002	14	100	0.002	11	73	0.12
	Per-Patient Analysis	11	5	45	11	100	0.01	11	100	0.01	9	82	0.18

**Table 3 jcm-10-04017-t003:** Impact of the WB-PET MRI on patient management compared to the standard of care CT-CAP/liver MRI combination.

Patient Number	Inclusion Criteria	BCLC	CT-CAP/Liver MRI Findings	CT-CAP/Liver MRI Metastatic Staging	WB-PET/MRI Findings	WB-PET/MRI Metastatic Staging	Final Allocated Treatment
26	Pre TARE workup	C	Nonspecific mediastinal lymph nodes	Not Metastatic	Hypermetabolic mediastinal lymph nodes	Metastatic	Palliative care
27	Pre liver transplantation workup	C	Indeterminate 5 mm size lung nodule	Probably metastatic	Lung nodule visible on DWI with hypermetabolism + 5 bone lesions on MRI (including 2 hypermetabolic lesions)	Metastatic	Palliative care
32	Pre TARE workup	C	No metastasis	Not Metastatic	One 9 mm size hypermetabolic sternal bone lesion visible on MRI	Metastatic	Metastasis-centered stereotaxic radiotherapy
37	Pre liver transplantation workup	C	No metastasis	Not Metastatic	2 bone lesions + 1 retroperitoneal hypermetabolic 8 mm size lymph node	Metastatic	Palliative care
79	Pre TARE workup	C	No metastasis	Not Metastatic	One 10 mm size hypermetabolic spinal bone lesion also visible on MRI	Metastatic	Metastasis-centered stereotaxic radiotherapy
95	Pre TARE workup	C	No metastasis	Not Metastatic	One 20 mm size hypermetabolic iliac bone lesion also visible on MRI	Metastatic	Metastasis-centered stereotaxic radiotherapy
97	Pre TARE workup	C	Nonspecific mediastinal lymph nodes	Not Metastatic	Hypermetabolic mediastinal lymph nodes	Metastatic	Palliative care
5	Pre surgery Workup	A	11 mm rounded lung nodule	Metastatic	No metastatic site. Lung nodule not visible on DWI and no hypermetabolic	Not Metastatic	Liver surgery
7	Pre surgery Workup	A	10 mm size lytic bone lesion	Metastatic	No metastatic site. Bone lesion typical of angioma on MRI and not hypermetabolic	Not Metastatic	Liver surgery
76	Pre surgery Workup	A	10 mm size left adrenal lesion >10 UH without signal fall on out phase	Metastatic	Not hypermetabolic left adrenal lesion	Not Metastatic	Liver surgery

## Data Availability

The data presented in this study are available on reasonable request from the corresponding author subject to approval by the research ethics committee of Henri Mondor hospital.

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
