# Peer review of "Impact of the 18F-FDG-PET/MRI on Metastatic Staging in Patients with Hepatocellular Carcinoma: Initial Results from 104 Patients"

_jcm, 2021, doi:10.3390/jcm10174017_

Round 1
Reviewer 1 Report
this is a well-conducted retrospective study comparing an innovative method (WB-PET / MRI) and a traditional one (CT-CAP / liver MRI) for the diagnosis of metastases in HCC patients eligible for curative / locoregional treatments. Some suggestions: 1) it would be useful to specify the CT image acquisition protocol. He is also always an experienced HCC radiologist to read the CT scan? While PET is usually done in a referral center for cancer, CT may not be. 2) the statistical significance value is missing in the comparison between sensitivity (100% vs 43%) between the 2 methods 3) an analysis of the different costs of the 2 methods and the calculation of the radiant dose could be useful 4) from table 2 it emerges that WB-MRI is the real advantage in terms of diagnosis of metastasis. Why is PET needed? it is useful to specify this in the discussion 5) specify the data on the false positives of the pet. It is useful to discuss the use of FDG rather than choline 6) I would give less emphasis to the conclusion and would emphasize more the limitations of the study (retrospective, low sample size, possible selection bias, limited follow up). This pilot study opens up some scenarios of potential scientific interest although some methodological corrections are necessary. It is important that it is integrated with a cost-benefit analysis and the calculation of the radiant dose between the 2 methods. the limits linked to the low sample size must also be emphasized more strongly
Reviewer 2 Report
The authors report that WB-PET/MRI may better detect HCC than CT-CAP/liver MRI combination. The study is relevant and interesting, it deserves to be published. However, I think that addressing the following issues would improve its quality.
- Follow up images are presented only for patient in Figure 4. Is it possible to add follow up images for Figures 2, 5 and 6, or to include them in supplementary material?
- For 90 patients found to be non-metastatic by both methods: is it possible that some of them had metastasis which was undetected by both methods? If I understood correctly 3 patients assessed non-metastatic by both methods died before the follow-up. Did they have a significant rise in serum alpha-fetoprotein, as did 2 metastatic patients described in the discussion? Please discuss
- The following section is not understandable to me: “The probability grading of metastasis on a per patient basis was identical on CT-CAP/liver MRI and WB-PET/MRI datasets (1.20 vs 1.15; p=0.23). However, in metastatic patients the mean probability of metastases was significantly higher when using WB-PET/MRI over standard of care imaging (3.64 vs 2.36).” How were these numbers and p-value obtained? Also, the second p value is missing.
- Is it possible that a single small size metastasis in the lungs wouldn’t be detected by WB-PET/MRI? Please discuss and add to possible limitations if necessary.
- In the discussion the sentence: “we found a 100% accuracy for WB-PET/MRI. There are several reasons for these results. First, only 15 metastatic sites in 11 patients were present in this patient population.“ should be moved to the next paragraph (limitations). Both paragraphs need to be revised.
- TMB abbreviation in abstract should be explained (or deleted?).
